# Unraveling the Role of the Zinc-Dependent Metalloproteinase/HTH-Xre Toxin/Antitoxin (TA) System of *Brucella abortus* in the Oxidative Stress Response: Insights into the Stress Response and Virulence

**DOI:** 10.3390/toxins15090536

**Published:** 2023-08-31

**Authors:** Leonardo A. Gómez, Raúl E. Molina, Rodrigo I. Soto, Manuel R. Flores, Roberto F. Coloma-Rivero, David A. Montero, Ángel A. Oñate

**Affiliations:** Laboratory of Molecular Immunology, Department of Microbiology, Faculty of Biological Sciences, University of Concepción, Concepción 4030000, Chile; ramolina@udec.cl (R.E.M.); rodrsoto@udec.cl (R.I.S.); manuelflores@udec.cl (M.R.F.); rcoloma@udec.cl (R.F.C.-R.); davmontero@udec.cl (D.A.M.)

**Keywords:** host-pathogen interaction, toxin-antitoxin (TA) systems, stress adaptation, virulence factors, transcriptional regulation

## Abstract

Toxin/antitoxin (TA) systems have been scarcely studied in *Brucella abortus*, the causative agent of brucellosis, which is one of the most prevalent zoonotic diseases worldwide. In this study, the roles of a putative type II TA system composed by a Zinc-dependent metalloproteinase (ZnMP) and a transcriptional regulator HTH-Xre were evaluated. The deletion of the open reading frame (ORF) BAB1_0270, coding for ZnMP, used to produce a mutant strain, allowed us to evaluate the survival and gene expression of *B. abortus* 2308 under oxidative conditions. Our results showed that the *B. abortus* mutant strain exhibited a significantly reduced capacity to survive under hydrogen peroxide-induced oxidative stress. Furthermore, this mutant strain showed a decreased expression of genes coding for catalase (*kat*E), alkyl hydroperoxide reductase (*ahp*C) and transcriptional regulators (*oxy*R and *oxy*R-like), as well as genes involved in the general stress response, *phy*R and *rpo*E1, when compared to the wild-type strain. These findings suggest that this type II ZnMP/HTH-Xre TA system is required by *B. abortus* to resist oxidative stress. Additionally, previous evidence has demonstrated that this ZnMP also participates in the acidic stress resistance and virulence of *B. abortus* 2308. Therefore, we propose a hypothetical regulatory function for this ZnMP/HTH-Xre TA system, providing insight into the stress response and its potential roles in the pathogenesis of *B. abortus*.

## 1. Introduction

Toxin–Antitoxin (TA) systems are small genetic elements widely distributed among plasmids and chromosomes of procaryotes [1,2]. These TA systems are composed of a toxin (usually a protein, with the exception of type VIII toxin), which inhibits an essential cellular process and an antitoxin (can be either a protein or a RNA), which neutralizes the catalytic function of the toxin [2,3]. Based on the mechanisms by which the antitoxin inactivates the toxin, these TA systems have been classified into eight types (I–VIII) [2,3]. Their functions in bacterial physiology are diverse and include the stabilization of mobile genetic systems, the inhibition of phages, inducing programed cell death (PCD), promoting biofilm formation and providing resistance to antibiotics and stress conditions [3,4]. These functions are associated with the release of the toxin, which occurs when the antitoxin is degraded during stressful conditions, allowing it to act on several targets, such as transcription, protein translation, cell wall synthesis, and/or inducing metabolic stress [3,5,6]. These phenomena arrest bacterial growth and can generate a bacterial dormancy or persister state, which are phenotypes less susceptible to stressors, such as antibiotics, acidic conditions or oxidative compounds [5]. These phenotypes allow bacteria to persist inside the host cell for long periods of time despite antibiotic treatments or other stressors [5]. Furthermore, it was demonstrated that TA systems are involved in bacterial virulence, being required for colonization, intracellular survival, and infection processes of diverse bacterial pathogens [7,8,9,10,11,12,13,14,15].

Many intracellular bacteria possess diverse types of TA systems, which could be involved in the resistance against oxidative and acidic stress, and in the formation of dormant or persister phenotypes. Nevertheless, these systems have been scarcely studied in *Brucella abortus*, the causative agent of brucellosis, one of the most prevalent zoonotic diseases worldwide [16]. For instance, the BrnT/BrnA type II TA system was described in *B. abortus* by Heaton et al. [17], which is constituted by a ribonuclease toxin with a RelE-like fold and an antitoxin BrnA. This BrnT/BrnA complex is induced by different environmental stressors, such as oxidative and acidic pH stress, where the BrnT released causes the arrest of bacterial growth. The activity of BrnT can be neutralized by the expression of the antitoxin BrnA [17]. Furthermore, a recent report showed a high expression of several TA systems in clinical *B. abortus* and *Brucella melitensis* strains isolated from animals and humans, including RelE/RHH-like, Fic/Phd, Brnt/BrnA and cogT/cogAT [18,19]. The expression of all these TA systems was increased by the presence of gentamicin; and all, with the exception of cogT/cogAT, were also increased under acidic conditions [19]. Interestingly, *B. abortus* is exposed to these oxidative and acidic conditions during its intracellular trafficking through endosomes and lysosomes in macrophages and neutrophils. These phagocytic cells use several mechanisms to eliminate microorganisms, such as reactive oxygen or nitrogen species (O_2_^−^, H_2_O_2_ and NO) and acidic pH are produced by the respiratory burst and the activity of vacuolar proton ATPases, respectively [16,20,21]. However, *B. abortus* has evolved diverse mechanisms of immune evasion and the capacity to survive intracellularly in these cells, including a *Brucella*-containing vacuole (BCV) trafficking for early endosomes (EE), late endosomes (LE) and lysosomes (lys) [16,20,21]. Principio del formuary. Although this intracellular environment is highly microbicidal for multiple microorganisms, oxidative and acidic conditions induce in *B. abortus* the expression of superoxide dismutase (SodC), catalase (KatE) and alkyl hydroperoxide reductase (AhpC), enzymes that protect the bacterium from these oxidant compounds [22]. Moreover, the acidification of the BCV induces the expression of the *virB* operon, a group of genes coding for the type IV secretion system (T4SS) VirB, in *B. abortus* [23]. This T4SS VirB system allows *B. abortus* to translocate multiple effector proteins, such as Btp1/BtpA, RicA or BspB into the cytoplasm of infected cells to inhibit intracellular signaling pathways and vesicular transport in host cells [24,25]. These translocated effectors enable *B. abortus* to escape from endosome–phagolysosome compartments to vesicles derived from the endoplasmic reticulum (ER), where it develops its intracellular replicative niche [16,24,25].

Sophisticated regulatory mechanisms are required by *B. abortus* to survive intracellularly and to complete its lifecycle in diverse types of host cells during the infection. These mechanisms include the expression of BvrR/BvrS, VjbR/BlxR or HutC, which are activated by acidic pH conditions and, subsequently, trigger the expression of T4SS VirB and the translocation of several effector proteins involved in the virulence of this bacterium [26,27,28,29]. Furthermore, under acidic and oxidative stress, the General Stress Response (GSR) is activated through factors PhyR and sigma RpoE1 (σ) [30]. This response is required by *B. abortus* to survive under stress conditions, and to induce a chronic infection in mammalian hosts [30]. This evidence demonstrates that several regulatory factors participate in the adaptation of *B. abortus* during its intracellular lifecycle in professional and non-professional phagocytic cells. Interestingly, our group described the roles of a zinc-dependent metalloproteinase (ZnMP), which forms an operon with an HTH-Xre regulator, constituting a putative type II TA system in *B. abortus* [31]. Results showed that deletion of ORF BAB1_0270 (ZnMP) significantly affected the capacity of this pathogen to survive under acidic conditions, to escape from phagolysosomes to reach its replicative niche in ER-derived vesicles, to survive within phagocytic cells and to colonize mice tissues [31,32]. Notably, the deletion of BAB1_0270 (ZnMP) significantly reduced the expression of various genes coding for transcriptional regulators (e.g., *vjbR*, *hutC*, *bvrR*), T4SS VirB proteins (e.g., *virB1*, *virb2*, *virB5*) and their effectors (e.g., *btpA*, *vceA* and *vceC*) [31]. Therefore, considering that these TA systems have been scarcely studied in *B. abortus,* this work aimed to evaluate the roles of the type II ZnMP/HTH-Xre TA system in the survival of *B. abortus* 2308 subjected to oxidative conditions. Based on previous results, and those obtained in this study, we are proposing a regulatory mechanism for this TA system during the stress response and the virulence of *B. abortus.*

## 2. Results

### 2.1. Characterization of Type II TA System in B. abortus Strains

The TA systems in *Brucella* species have been poorly studied. In this study, a bioinformatic search was conducted to identify these systems in *B. abortus* 2308 and 9-941 strains. A search in the Toxin-Antitoxin Database TADB2.0 [33] allowed us to identify at least four potential type II TA systems present in the chromosome I of *B. abortus* biovar 1 strain 9-941 (https://www.ncbi.nlm.nih.gov/nuccore/NC_006932.1, accessed on 29 August 2023) and *B. abortus* strain 2308 (https://www.ncbi.nlm.nih.gov/nuccore/NC_007618, accessed on 29 August 2023) (Table 1). One of these TA systems is composed of the ZnMP (toxin) coded in the ORF BAB1_0270 (WP_002965518.1) present in a genomic island called “genomic island 3”, which is shared by some *Brucella* species, such as *B. abortus* 2308 and 9-941 strains [31,32,34]. This protein is homologous to the ImmA/IrrE metalloproteinases family, which is characterized by conserved domains COG2856 and motif HEXXH (pfam06114). Furthermore, this ZnMP forms an operon with a transcriptional-Helix-Turn-Helix (HTH)-Xre (Xenobiotic response element) regulator, which is constituted by a nucleotide sequence of 357 bp (WP_002967122.1) that translates a 118 amino acids (aa) protein (pfam13443) (Appendix A). Thus, both genes form an operon located in the region 270,612–271,513 of chromosome I in *B. abortus* 2308 (NC_007618.1) [31], similar to the operon coded by the ORFs BruAb1_0264 and BruAb1_0263 of *B. abortus* biovar 1 strain 9-941, which has been described by TADB2.0 as a type II TA system. Therefore, all evidence supports idea that ZnMP and HTH-Xre transcriptional regulator constitute a type II TA system, where ZnMP is the toxin and HTH-Xre is the antitoxin (Table 1).

### 2.2. ZnMP Is Required in the Resistance of B. abortus against Oxidative Conditions

Type II TA systems play significant roles in the bacterial adaptation to stress induced by acidic and/or oxidative conditions. The role of ZnMP protein in the stress resistance of *B. abortus* strains under oxidative conditions induced by various concentrations of H_2_O_2_ was measured. The results showed that the viability of the three strains was initially similar under low concentrations of H_2_O_2_ (from 0.1 to 0.2 mM) (*p* > 0.5); however, at concentrations of 0.4, 0.6 and 0.8 mM H_2_O_2_, the viability of *B. abortus* 270 strain (mutant for the ORF BAB1_0270) was significantly reduced when compared to the wild-type strain (*p* < 0.01) or the complemented strain *B. abortus* 270C (*p* < 0.01) (Figure 1). Moreover, H_2_O_2_ concentrations from 1.0 to 1.2 mM severely reduced the viability of *B. abortus* 270 strain when compared to the wild-type and complemented strains (Figure 1). These results indicate that type II TA system based on ZnMP/HTH-Xre is required for *B. abortus* to resist or tolerate oxidative conditions.

### 2.3. ZnMP (COG2856) in the Expression of Genes Coding for Antioxidant Components in B. abortus 2308

To survive inside phagocytic cells, *B. abortus* expresses diverse types of enzymes, such as catalase and alkyl hydroperoxide reductase C. The expression of these enzymes is mediated by OxyR transcriptional factors, as illustrated in the model described in Figure 2A,B. The expression of *kat*E (ORF BAB2_0849), *ahp*C (ORF BAB2_0531), and the potential regulators *oxy*R (BAB2_0849) and *oxy*R-like (BAB2_0530) genes was measured by RT-qPCR. Results showed that *B. abortus* 2308 cultured under oxidative conditions upregulates the expression of *kat*E (*p* < 0.05) and *ahp*C (*p* < 0.05) when compared to *B. abortus* 270 (Figure 2C,D). Furthermore, the expression of *oxy*R and oxyR-like *lys*R regulators was also significantly increased in *B. abortus* 2308 when compared to *B. abortus* 270 (Figure 2E,F) when these strains were subjected to oxidative conditions (*p* < 0.05 and *p* < 0.01, respectively). Nevertheless, although the expression of *kat*E and *ahp*C genes in *B. abortus* 270 was significantly reduced when compared to wild-type strain, their expression was slightly less or similar, respectively, when it was compared to control group (Figure 2C,D). Similarly, the expression of the transcriptional regulators *oxy*R and OxyR-like *lys*R in *B. abortus* 270 was slightly less than the control group (Figure 2E,F). Complemented strain *B. abortus* 270C showed higher levels of expression for *kat*E, *ahp*C, *oxy*R and oxyR-like *lys*R regulators when compared to *B. abortus* 270 (Appendix A). These results demonstrate that the deletion of ZnMP negatively affects the expression of several genes involved in the survival of *B. abortus* under oxidative stress conditions.

### 2.4. ZnMP in the Expression of phyR and rpoE1 of B. abortus

*B. abortus* requires the general stress response (GSR) to survive under acidic and oxidative conditions. This response is regulated by several components, such as PhyR and RpoE1, which are illustrated in the model shown in Figure 3A. The effects, by deleting ZnMP, in the expression of *phy*R (BAB1_1671) and *rpo*E1 (BAB1_1672) genes in *B. abortus* strains exposed under oxidative conditions was evaluated by RT-qPCR. The results showed that *phy*R and *rpo*E1 were significantly upregulated in *B. abortus* 2308 exposed to oxidative conditions (*p* < 0.01 and *p* < 0.05, respectively) compared to *B. abortus* 270 (Figure 3B,C). However, in *B. abortus* 270, the expression of *phy*R was slightly lower than control group, while the expression of *rpo*E1 was similar to the control group. Complemented strain *B. abortus* 270C showed higher levels of expression for *phyR* and *rpo*E1 when compared to *B. abortus* 270 (Appendix A). These genes coding for PhyR, a repressor of NepR, and for a RNA polymerase sigma factor (σ), respectively, are involved in the GSR of *B. abortus*. Therefore, these results suggest that type II ZnMP/HTH-Xre TA system contributes in the oxidative stress response of *B. abortus.*

## 3. Discussion

The type II TA systems are operons found in many bacterial and plasmid genomes that encode a toxin and an antitoxin protein [1,2,3]. These genetic module physically form a toxin–antitoxin complex, where the antitoxin inhibits the toxin’s activity on its cellular targets [1,2,3]. Toxins released can affect diverse bacterial functions, such as plasmid post-segregational killing, programmed cell death, defense against phages, biofilm formation, as well as antibiotic-induced persistence, or resistance to stress conditions [1,6]. These roles are associated with the inhibition of bacterial growth and/or induction of dormant or persister states, which are phenotypes less susceptible to stressors, such as antibiotics, acidic or oxidative compounds [5]. Recently, we demonstrated that ZnMP (BAB1_0270) of *B. abortus* strain 2308 forms an operon with a HTH-Xre transcriptional regulator, which may be a type II TA system [31]. By RT-PCR assays, it was confirmed that the messenger RNA (mRNA) of this *B. abortus* 2308 operon amplifies a PCR product of 906 pb containing the ORF BAB1_0270 of 549 bp (sequence for ZnMP) and the HTH-Xre transcriptional regulator of 357 bp [31]. In addition, this operon has a promoter in the regions -35 and -10 before HTH-Xre/ZnMP sequence, which is recognized by RpoD sigma factors of the RNA polymerase [31]. The deletion of ZnMP significantly reduced the growth of *B. abortus* lacking this protein under acidic conditions, the expression of several genes involved in the virulence and, consequently, reduced its ability to survive in macrophages and to colonize tissues in BALB/c mice [31,32]. Moreover, the deletion of ZnMP significantly reduced the ability of *B. abortus* to adapt intracellularly and to grow under acidic pH [31]. Interestingly, this deletion did not affect the growth of the mutant strain when compared to wild-type strain under physiological conditions (pH 7.2–7.4) [32]. Therefore, in this work, we studied the roles of this type II ZnMP/HTH-Xre TA system in the survival and gene expression of *B. abortus* 2308 cultured under oxidative conditions with the aim to understand whether this TA system exerts regulatory functions during the stress response of *B. abortus* 2308.

Our results revealed that *B. abortus* 2308 has four putative type II TA systems in chromosome I, including COG2856(ZnMP)/HTH XRE, RelE/RHH, Phd-Doc/AbrB, and BrnT/BrnA (Table 1). The ZnMP/HTH-Xre is an operon that consists of an Imma/IrrE family metallopeptidase (ZnMP) and a HTH-Xre transcriptional regulator [31]. It is a pair gene (Xre-COG2856) similar to those described in the related *B. abortus* biovar 1 strain 9-041 (Table 1) and also present in phylogenetically distant Gram-positive *Streptococcus pneumoniae* strains and *Clostridioides difficile*, as well as in bacterial thermophiles *Geobacillus* spp. [35,36,37]. These genes coding for COG2856/Xre are often found in mobile genetic elements, such as bacteriophages or transposons. Furthermore, COG2856 (ZnMP) proteins contain a conserved HEXXH motif, which is commonly associated with proteins that contain the HTH domain of the Xre family. In *Bacillus subtilis*, these proteins are involved in the genetic control of bacterial cell death, specifically in the induction of late genes of prophage PBSX, which largely depend on the *xre* gene [38]. This Xre protein regulates not only its own expression, but also the expression of other genes, including a downstream regulatory complex cascade [38]. On the other hand, ZnMP (COG2856) shares homology with proteins of the ImmA and IrrE families, which are present in *B. subtilis* and *Deinococcus radiodurans* and *Deinococcus deserti*, respectively [39,40,41]. ImmA and IrrE are proteases that cleave repressor proteins, such as ImmR and Ddro, respectively. Thus, ImmA cleaves ImmR, favoring the expression of genes involved in the transfer of mobile genetic elements, such as ICEBs1, and IrrE cleaves Ddro, favoring the transcription of several genes required for bacterial survival under conditions of ionizing radiation or ultraviolet radiation [39,40,41,42,43]. Moreover, IrrE is a key regulator of the recA gene [39,42,43,44], a protein strongly induced during irradiation, which orchestrates the cellular response to DNA damage via its multiple roles in the SOS response [45,46,47]. It is possible that the role played by IrrE in other bacteria may be performed by ZnMP in *B. abortus*, suggesting that this protein might exert pleiotropic effects that control large sets of genes encoding different metabolic and signaling pathways [42]. Additionally, some authors indicate that these COG2856/Xre proteins could be analogues of the well-characterized SOS response regulator LexA, a repressor of complex regulons [48,49,50]. These regulatory functions of IrrE have been demonstrated by transforming *Escherichia coli* and other bacteria with recombinant plasmids encoding the IrrE protein. This transformation has improved the tolerance of these bacteria to various abiotic stresses, such as irradiation, osmotic changes, heat, salt, and oxidative stress [51,52,53,54].

The roles of ZnMP under oxidative stress induced by H_2_O_2_ showed, by deleting ZnMP, a significant impact on the growth of *B. abortus* 270 (Figure 1). This effect was associated with a lower expression of genes coding for catalase (BAB2_0848) and AhpC (BAB2_0531), key enzymes in the survival of intracellular pathogens, such as *Brucella* species, *Mycobacterium tuberculosis* or *Burkholderia* spp. These enzymes detoxify peroxides, organic peroxides, and peroxynitrites into less toxic elements such as O_2_ and H_2_O_2_ [2,54,55,56,57,58,59]. Interestingly, the lower expression levels of antioxidant enzymes were directly associated with a lesser expression of *oxy*R (BAB1_0849) and *oxy*R-like (BAB2_0530), which are genes adjacent to *kat*E (BAB2_0848) and *ahp*C (BAB2_0531), respectively, and that could be transcriptional regulators of these enzymes during bacterial exposition to H_2_O_2_ (Figure 2A,B) [60]. Recently, it was demonstrated that OxyR-like (BAB2_0530) is involved in the resistance of *B. abortus* to nitrosative stress, detergents, and also virulence, regulating diverse pathways, including nitrogen metabolism, siderophore biosynthesis and oligopeptide transport [61]. On the other hand, the expression of *phy*R and *rpo*E1 also was significantly reduced in *B. abortus* 270 when compared to the wild-type strain. These elements are involved in the GSR of *B. abortus*, which is a system consisting of a phospho-LovhK (LovhK~P) that transfers its phosphoryl group to PhyR (~P), which increases its affinity for NepR, a repressor of RpoE1 σ factor (Figure 3A) [31]. The binding between PhyR~P to NepR allows to release the σ-factor (RpoE1), which can associate the RNA polymerase (RNAP) to a specific genomic region regulating the transcription of genes required by the bacterium to survive under acute stress conditions [30,62,63]. Although the relationship between RpoE1, catalase, AhpC, OxyR and OxyR-like regulators is still unknown, it has been demonstrated in *Azospirillum brasilense* that RpoE1 can regulate the gene expression of OxyR and catalase [64,65]. These findings reporting the changes in gene expression could explain why a low expression of antioxidant response was observed in *B. abortus* 270 mutant strain.

Therefore, the evidence suggests that ZnMP plays critical roles in stress survival of *B. abortus*. Thus, the release of ZnMP from the toxin–antitoxin complex could allow it to exert its protease activity, leading to the development of dormant or persister phenotypes [5]. Nevertheless, considering the homology between ZnMP and IrrE protein, we hypothesize that ZnMP exerts a similar function in *B. abortus,* regulating pleiotropically diverse mechanisms involved in the stress response and virulence [31]. Thus, we propose a model to explain the complex roles that this type II ZnMP/HTH-Xre TA system could play in the physiology of *B. abortus* (Figure 4A,B). In this model, during the stress response, the type II toxin ZnMP is released from the antitoxin HTH-Xre by the activity of intracellular proteases, cleaving repressor proteins (e.g., HTH-Xre proteins) that inhibit the expression of genes involved in the stress response and virulence of *B. abortus*. These regulatory functions could represent a new pathway for type II TA systems, in which the released toxin does not induce toxic effects, such as cellular death, but rather controls the transcriptional expression of several genes involved in the stress response and/or virulence. Therefore, this hypothetical function challenges the historical conception about the functions of these TA systems, where they could accomplish more complex regulatory circuits than those currently described for type II TA systems. Additional experimental analyses are required to validate that the TA system here studied in silico accomplishes the specific regulatory functions to understand its contribution to the fitness of *B. abortus* during the infection. This understanding may help us to develop new strategies for the treatment of brucellosis.

## 4. Conclusions

In silico analysis demonstrated that Zinc-dependent metalloproteinase (ZnMP) (toxin) constitutes a type II TA system with a HTH-Xre transcriptional factor (antitoxin). ZnMP plays a crucial role in *B. abortus*´ adaptation to oxidative stress. Its deletion reduced the ability of *B. abortus* to survive when subjected to the presence of H_2_O_2_ and it also reduced the expression of antioxidant enzymes or regulatory factors involved in the response against oxidative stress when compared to the wild-type strain. Based on the evidence reported here, and previous reports demonstrating that ZnMP is required for *B. abortus* virulence, we can conclude that this postulated TA system is implicated in the regulation of the stress response and virulence of this bacterium. Therefore, this hypothetical function of ZnMP/Xre TA system could provide new insights into the regulatory mechanisms underlying the virulence and the stress response of *B. abortus*, thereby expanding our understanding on the roles of TA systems and how this pathogen adapts to its intracellular niche, a crucial adaptation in the progression of brucellosis in animals and humans.

## 5. Materials and Methods

### 5.1. Prediction of Type II TAs Loci in Chromosome I of B. abortus

Type II TA system present in the chromosome I of *B. abortus biovar* 1 strains 9-941 and *B. abortus* 2308 were analyzed using the Toxin–Antitoxin Database (TADB2) and TAfinder for type II TA system (http://bioinfo-mml.sjtu.edu.cn/TADB2/, accessed on 29 August 2023) [33]. Furthermore, comparative analysis for ImmA/IrrE family metallo-endopeptidase (WP_002965518.1) and HTH-Xre transcriptional regulator (WP_002967122.1) elements of type II Tas, were analyzed using Basic Local Alignment Search Tool (BLAST, https://blast.ncbi.nlm.nih.gov/Blast.cgi, accessed on 29 August 2023) from the GenBank database (https://www.ncbi.nlm.nih.gov/nuccore/NC_007618, accessed on 29 August 2023).

### 5.2. Bacterial Strains and Culture Conditions

*B. abortus* 2308 (wild type strain), *B. abortus* mutant strain for ORF BAB1_0270 (called *B. abortus* 270) and *B. abortus* complemented strain for BAB1_0270 (called *B. abortus* 270C strain) were used in this work [32]. All strains were cultured in Brucella broth (BD Difco, Thermo Fisher Scientific, Waltham, MA, USA) and incubated at 37 °C for 48–72 h. Additionally, *B. abortus* 270 and *B. abortus* 270C were cultured in Brucella broth with 50 μg/mL kanamycin and 30 μg/mL ampicillin, respectively, according to Ortiz-Román et al. [32]. All assays and experiments were performed following the procedure stablished by the Biosafety Committee of the University of Concepcion (Concepcion, Chile).

### 5.3. Viability of B. abortus under Oxidative Conditions

The role of ZnMP in the resistance of *B. abortus* to the stress when subjected to oxidative conditions was analyzed by exposing *B. abortus* 2308, *B. abortus* 270 and *B. abortus* 270C strains to 0, 0.2, 0.4, 0.6, 0.8, 1.0 or 1.2 mM hydrogen peroxide (H_2_O_2_) (Merk, Darmstadt, Germany) for 20 min. The viability of these bacterial strains was quantified by a colorimetric assay based on the use of the PrestoBlue reagent according to the manufacturer’s protocol (Thermo Fisher Scientific Inc., Waltham, MA, USA). Absorbance was measured on a VictorX3 microplate reader (PerkinElmer, Waltham, MA, USA) at 590 nm. Bacterial survival was analyzed in triplicate and results were expressed as the relative fluorescence units mean ± standard deviation.

### 5.4. Gene Expression of B. abortus Strains under Oxidative Conditions

Regulatory functions of ZnMP when this bacterium was subjected to oxidative stress was analyzed by gene expression of antioxidant components, such as catalase *kat*E (ORF BAB2_0849), alkyl hydroperoxide reductase *ahp*C (ORF BAB2_0531), *oxy*R transcription factor (ORF BAB2_0848), *oxy*R-like LysR family transcription factor (ORF BAB2_0530), and components of the general stress response, such as *phy*R (BAB1_1671) and *rpo*E1 (BAB1_1672) through RT-qPCR. Briefly, *B. abortus* strains were cultured in the presence of 0.5 mM H_2_O_2_ during 24 h at 37 °C in Brucella broth. Later, total RNA was extracted from the pellets of each *B. abortus* strain, and centrifuged at 7500× *g*, using TRIzol (Thermo Fisher Scientific Inc., Waltham, MA, USA) following the instructions of the manufacturer. RNA was transformed to complementary DNA (cDNA) by reverse transcription using the Maxima First Strand cDNA Synthesis kit for RT-PCR (Thermo Fisher Scientific Inc., Waltham, MA, USA) and concentration of cDNA was adjusted using a Tecan Infinite M Nano equipment (Tecan Trading AG, Switzerland) to analyze the relative expression of the genes of interest (Table 2). The expression of each gene was quantified by the 2^−ΔΔ*CT*^ method using the Takyon q-PCR kit for SYBR assays by means of the AriaMx Real Time PCR system (Agilent Technologies, Santa Clara, CA, USA). The *16s* housekeeping gene was used as reference gene for all assays. Additionally, total RNA from *B. abortus* 2308 wild-type strain cultured under normal conditions was used as a control group for gene expression (value 1). The gene expression was analyzed in triplicate and the results were expressed as the mean ± standard deviation.

### 5.5. Statistical Analysis

The statistical differences in the survival among strains cultured under oxidative conditions was determined by a two-way ANOVA. Gene expression was analyzed by non-parametric Mann–Whitney tests. All statistical analyses were performed using GraphPad Prism 9 software (GraphPad Software, San Diego, CA, USA). *p* values < 0.05 were considered as statistically significant.

## Figures and Tables

**Figure 1 toxins-15-00536-f001:**
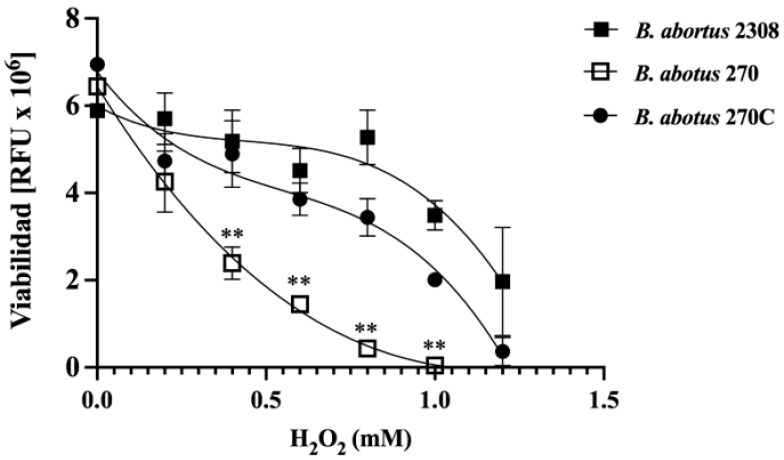
Role of ZnMP in the resistance of *B. abortus* under oxidative conditions. *B. abortus* 2308, *B. abortus* 270 and complemented *B. abortus* 270C strains were subjected to 0, 0.2, 0.4, 0.6, 0.8, 1.0, 1.2 mM hydrogen peroxide for 20 min. The bacterial viability of each strain was quantified by a colorimetric assay measuring the relative fluorescence units (RFU) at 590 nm using PrestoBlue reagent (resazurin). Results are shown as the mean ± standard deviation. *p* values < 0.05 were considered statistically significant, ** denotes *p* values < 0.01. All tests were performed in triplicate.

**Figure 2 toxins-15-00536-f002:**
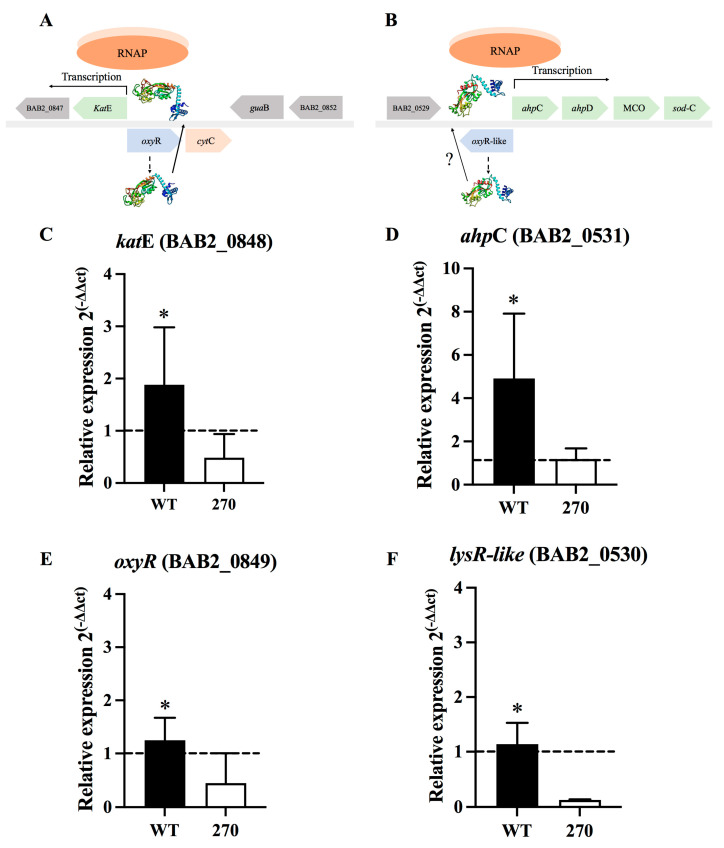
Deletion of ZnMP in the expression of genes coding for enzymes and regulators involved in the resistance of *B. abortus* under oxidative conditions. Model for transcriptional regulation of (**A**) catalase *kat*E (ORF BAB2_0849) by OxyR transcription factors (ORF BAB2_0848) and (**B**) alkyl hydroperoxide reductase *ahp*C (ORF BAB2_0531) by OxyR-like LysR family transcription factors (ORF BAB2_0530) in *B. abortus* 2308. Gene expression of (**C**) *kat*E, (**D**) *ahp*C, (**E**) *oxy*R and (**F**) *oxy*R-like. The expression of these genes was analyzed in *B. abortus* 2308 and in *B. abortus* 270 mutant for ZnMP cultured for 24 h under oxidative stress induced by H_2_O_2_. *B. abortus* 2308 grown in culture medium without H_2_O_2_ was used as a control group for gene expression (no induction, value 1). The expression of these genes was calculated by the 2^−ΔΔCT^ method using RT-qPCR. The *16s* housekeeping gene was used as reference. Results were expressed as the mean ± standard deviation. * denotes *p* value < 0.05 was considered statistically significant. ? indicates a possible role of OxyR-like in the transcriptional regulation of AhpC. All tests were performed in triplicate. Abbrev. RNAP: RNA polymerase, MCO: Multi cooper oxidase.

**Figure 3 toxins-15-00536-f003:**
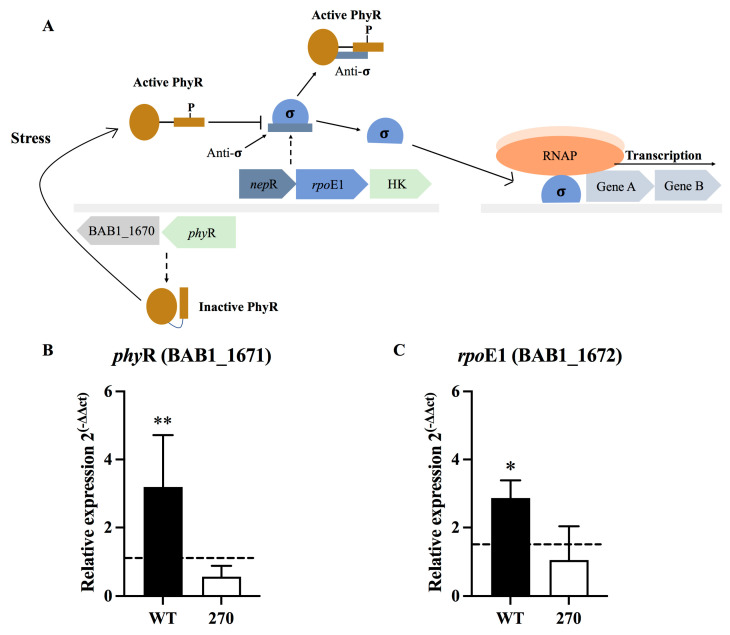
Deletion of ZnMP in the general stress response of *B. abortus*. (**A**) Model for activation of general stress response and its components. (**B**) Expression of *phy*R gene (ORF BAB1_1671) coding for anti-anti-repressor nepR. (**C**) Expression of *rpo*E1 gene (ORF BAB1_1672) coding for sigma factor (σ). The expression of these genes in *B. abortus* 2308 and *B. abortus* 270 mutant for ZnMP cultured for 24 h under oxidative stress induced by H_2_O_2_. *B. abortus* 2308 grown in culture medium without H_2_O_2_ was used as a control group for gene expression (no induction, value 1). The expression of these genes was calculated by the 2^−ΔΔCT^ method using RT-qPCR. The *16s* housekeeping gene was used as reference. Results were expressed as the mean ± standard deviation. A *p* value < 0.05 was considered statistically significant. All tests were performed in triplicate. * and ** denotes *p* values < 0.05 and 0.01, respectively.

**Figure 4 toxins-15-00536-f004:**
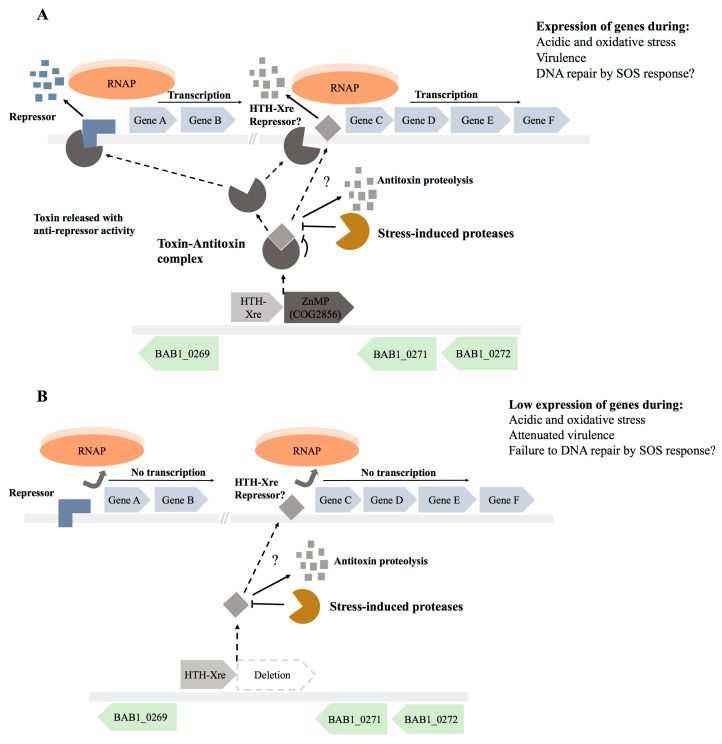
A hypothetical model for regulatory mechanisms mediated by anti-repressor activity of ZnMP during the stress response and virulence of *B. abortus*. (**A**) Expression of TA complex (ZnMP/HTH-Xre) in *B. abortus* 2308 under stress conditions leads the release of the ZnMP toxin by antitoxin proteolysis. The free ZnMP then exerts its proteolytic function against unknown repressors (possibly HTH-Xre repressor), enabling the transcription of several genes involved in the stress response, virulence and potentially in the DNA damage repair, such as the SOS-response; (**B**) The deletion of ZnMP in *B. abortus* 270 cultured under stress conditions prevents the cleavage of unknown repressors by ZnMP protease. As a result, RNA polymerase (RNAP) cannot bind to promoters, leading to a reduced transcription of several genes necessary for the survival of the bacterium under stress conditions. Dotted arrows and question marks (?) indicates that toxin may require various steps to reach its DNA target.

**Table 1 toxins-15-00536-t001:** Putative type II TAs present in the chromosome I of *B. abortus biovar* 1 str. 9-941 and *B. abortus* 2308 analyzed by TADB2.

	Biovar	Type	Toxin	Antitoxin	Location (Chr I)	Classification (T/A)
1	*B. abortus* bv. 1 str. 9-941	II	BruAb1_0264	BruAb1_0263	274,246–275,147	COG2856/HTH Xre
	*B. abortus* 2308	II	BAB1_0270	*BAB_RS17200	270,612–271,513	COG2856 (ZnMP)/HTH Xre
2	*B. abortus* bv. 1 str. 9-941	II	BruAb1_0430	BruAb1_0431	435,226–435,800	RelE/RHH-RelE
	*B. abortus* 2308	II	BAB1_0436	BAB1_0437	431,593–432,167	ParE/RHH-RelE
3	*B. abortus* bv. 1 str. 9-941	II	BruAb1_0579	BruAb1_0580	574,332–574,948	phd-doc/AbrB-Fic
	*B. abortus* 2308	II	BAB1_0581	BAB1_0582	570,627–571,243	Phd-doc/AbrB:SpoVT
4	*B. abortus* bv. 1 str. 9-941	II	BruAb1_0981	BruAb1_0980	962,125–962,759	BrnT/BrnA
	*B. abortus* 2308	II	BAB1_0994	BAB1_0993	962,125–962,759	BrnT/BrnA

*BAB_RS17200 is a new locus not previously described in the genome of *B. abortus* biovar 2308, which is similar to BruAb1_0263 from *B. abortus* biovar 1 str. 9-941. It codifies a transcriptional regulator present in *Brucella* and *Ochrobactrum* groups (WP_002967122.1).

**Table 2 toxins-15-00536-t002:** Primers for genes expression measured by RT-qPCR in *B. abortus* strains exposed to hydrogen peroxide.

Function	ORF/Gene		Sequence from 5′ to 3′
Housekeeping	16s	Forward	agctagttggtggggtaaagg
		Reverse	gctgatcatcctctcagacca
Antioxidant enzymes	*kat*E (BAB2_0848)	Forward	accatgggtgacgttcctc
		Reverse	gatgagcttcaggttcagca
	*ahp*C (BAB2_0531)	Forward	cgatctggtcgtttgctgata
		Reverse	caacgaaggtgtagcgatagg
Transcriptional factors	*oxy*R (BAB2_0849)	Forward	cttttcgacgaccgttttct
		Reverse	cgaggcgagaaccgtatg
	*oxy*R-like (BAB2_0530)	Forward	tactcgaaaccgggcatt
		Reverse	ctcatttgcgcaggcttt
General stress response	*phy*R (BAB1_1671)	Forward	attttcatcaccgcatttcc
		Reverse	cggcttggtgacgagaaa
	*rpo*E1 (BAB1_1672)	Forward	gcttggctcttcaccattct
		Reverse	agctgttcgctgaacatacc

## Data Availability

All experimental data that support the findings of this study are available from the corresponding author upon request.

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
