# Peer review of "Unraveling the Role of the Zinc-Dependent Metalloproteinase/HTH-Xre Toxin/Antitoxin (TA) System of Brucella abortus in the Oxidative Stress Response: Insights into the Stress Response and Virulence"

_toxins, 2023, doi:10.3390/toxins15090536_

Round 1
Reviewer 1 Report
The manuscript entitled “Unraveling the role of zinc-dependent metalloproteinase/HTH Xre toxin/antitoxin (TA) system of Brucella abortus in the oxidative stress response: insights into stress response and virulence” deals with the characterization of ZnMP/HTH-Xre TA system, which shows resistance against oxidative stress. The authors have revealed the function of this system and assesed against varying concentrations of expressions. The manuscript is well-written, and the discussion are supported by data. Also, literature citation is adequate and figures and tables are mostly suitable for the readers. therefore, only some minor points should be addressed.
Point 1:
Figures and tables: Figure 1 can be smaller. Also, table 1 is relatively big and over the size.
Point 2:
Additional conclusion can be added to emphasize the results.
Author Response
Dear Reviewer,
Thank you for your comments.
According your suggest for the point 1, the figure 1 and Table 1 were modified, reducing the size for each one.
Point 2, additional information was added in section “4. Conclusions” in lines 318 to 330 for emphasizing results: “In silico analysis demonstrated that Zinc-dependent metalloproteinase (ZnMP) (toxin) constitutes a type II TA system with a HTH-Xre transcriptional factor (antitoxin). ZnMP plays a crucial role in B. abortus´s adaptation to oxidative stress. Its deletion reduced the ability of B. abortus to survive when subjected to the presence of H2O2 and it also reduced the expression of antioxidant enzymes or regulatory factors involved in the response against oxidative stress when compared to the wild type strain. Based on the evidence here reported and previously reports demonstrating that ZnMP is required for B. abortus virulence, we can conclude that this postulated TA system is implicated in the regulation of the stress response and virulence of this bacterium. Therefore, this hypothetical function of ZnMP/Xre TA system could provide new insights into the regulatory mechanisms underlying the virulence and the stress response of B. abortus; thereby, expanding our understanding on the roles of TA systems and how this pathogen adapts to its intracellular niche, a crucial adaptation in the progression of brucellosis in animals and humans.”
With these changes, we hope to address your concerns regarding our work.
Best regards
Reviewer 2 Report
This paper is really interesting in the description of transcriptional regulation due to TA system of genes involved in the bacterial response to stresses.
The data are consistent with the proposed model, the only additional experiment I would suggest is to measure the relative level of expression of stress genes also for the complemented strain (strain 270C) to directly demonstrate that re-expression of ZnMP drives recovery of the production of stress genes
Only minor spelling control required
Author Response
Dear Reviewer,
Thank you for your comments.
According your suggest, we added new supplementary data for comparative PCR analysis between mutant and complementary strain was added (Figure Supplementary 1). These results show a higher expression for interest genes in complemented strain when compared to mutant strain. For this comparative analysis complemented strain was compared to mutant strain (Value 1).
Secondly, English spelling was reviewed and modified to better it in the following lines:
In line 36, “acts” was changed by “act”
In line 42, “involved in the bacterial virulence” was changed by “involved in bacterial virulence”
In line 47, “being required in the colonization..” was changed by “being they required for colonization…”
In line 65, “these cells” was changed by “these phagocytic cells”
In line 73, “enzymes that protect it” was changed by “enzymes that protect the bacterium”.
In line 103, “our work aims to evaluate” was changed by “this work aimed to evaluate…”
In line 113 was added “the” chromosome I.
In line 154, The phrase “Moreover, H2O2 concentrations from 1.0 to 1.2 mM completely reduced the viability” was changed by “ Moreover, H2O2 concentrations from 1.0 to 1.2 mM severely reduced the viability”
In line 166, “B. abortus expresses diverse types of enzymes, such as catalase and alkyl hydroperoxide reductase C, to survive inside phagocytic cells” was modified by “To survive inside phagocytic cells, B. abortus expresses diverse types of enzymes, such as catalase and alkyl hydroperoxide reductase C”.
In line 172, 174 and 177, was added “when” in the phrase “when compared to”
In line 175, “exposed” was changed by “subjected”
In line 208, “PhyR and RpoE1, which are illustrated by the model described in Figure 3A” was changed by “which are illustrated in the model shown in Figure 3A”
In line 373, “the” was added to “analyzed using the Toxin-Antitoxin Database)
In line 378, database was added to “using the GenBank database”
In line 398, “Regulatory functions of ZnMP during the exposure of this bacterium to oxidative stress was..” was changed by “Regulatory functions of ZnMP when this bacterium was subjected to oxidative stress was..”
In line 403, “B. abortus strains were cultured with 0.5 mM H2O2“ was changed by “B. abortus strains were cultured in the presence of 0.5 mM H2O2“
In line 405, “and centrifuged to 7500 x g” was changed by “and centrifuged at 7500 x g”
In line 417, “using Tecan Infinite M Nano” was changed by “using a Tecan Infinite M Nano equipment”
In line 320, “enzymes key in the survival” was changed by “key enzymes in the survival”
In line 336, “genes required for acute stress survival” was changed by “genes required by the bacterium to survive under acute stress conditions”
In line 340, “These findings could explain why” was changed by “These findings reporting the changes in gene expression could explain why”
In line 56, “where them could exert more complex regulatory circuit” was changed by “where they could accomplish more complex regulatory circuits”
Furthermore, Section abstract was reviewed and section “4. Conclusions” was added new information: “In silico analysis demonstrated that Zinc-dependent metalloproteinase (ZnMP) (toxin) constitutes a type II TA system with a HTH-Xre transcriptional factor (antitoxin). ZnMP plays a crucial role in B. abortus´s adaptation to oxidative stress. Its deletion reduced the ability of B. abortus to survive when subjected to the presence of H2O2 and it also reduced the expression of antioxidant enzymes or regulatory factors involved in the response against oxidative stress when compared to the wild type strain. Based on the evidence here reported and previously reports demosnstrating that ZnMP is required for B. abortus virulence, we can conclude that this postulated TA system is implicated in the regulation of the stress response and virulence of this bacterium. Therefore, this hypothetical function of ZnMP/Xre TA system could provide new insights into the regulatory mechanisms underlying the virulence and the stress response of B. abortus; thereby, expanding our understanding on the roles of TA systems and how this pathogen adapts to its intracellular niche, a crucial adaptation in the progression of brucellosis in animals and humans.”
With these changes, we hope to address your concerns regarding our work.
Best regards
Reviewer 3 Report
It is important to identify the physiological functions of TA systems in causative pathogens including Brucella abortus. The authors explored the functions of a putative type II TA system ZnMP/HTH-Xre and found its regulation of bacterial viability under different oxidative stress, and its regulation of expression of some enzymes involved in resistance, results mainly come from viability and qPCR.
The current data is quite expected and reasonable, and can basically support the conclusion that the putative toxin is important for adaptation to stress and regulation of gene expression, but, considering that the authors gave the massive physiological inference of ZnMP/HTH-Xre, additional data is quite necessary for this manuscript, in other words, current data is not so strong, following are two main suggestions:
1. it is necessary to directly demonstrate that BAB1_0270/BAB_RS17200 is a TA pair (putative toxin is toxic, putative antitoxin can neutralize).
2. it is unsuitable to evade the possible function of BAB_RS17200, which means you need to include a TA mutant and a TA complemented strain in your design
Author Response
Dear Reviewer,
Thank you for your comments.
We understand that in order to validate this TA system as you have requested, we need to develop a mutant strain for BAB_RS17200, and for the operon BAB_RS17200-BAB1_0270. However, we have not yet developed these mutants, which will require various months of work. For this reason, we have modified part of the discussion, indicating that " Additional experimental analyses are required to validate that the TA system here studied in silico accomplishes the specific regulatory functions to understand its contribution to the fitness of B. abortus during the infection." Furthermore, we have re-written the conclusion, highlighting the role of ZnMP rather than the TA system itself: "In silico analysis demonstrated that Zinc-dependent metalloproteinase (ZnMP) (toxin) constitutes a type II TA system with a HTH-Xre transcriptional factor (antitoxin). ZnMP plays a crucial role in B. abortus´s adaptation to oxidative stress. Its deletion reduced the ability of B. abortus to survive when subjected to the presence of H2O2 and it also reduced the expression of antioxidant enzymes or regulatory factors involved in the response against oxidative stress when compared to the wild type strain. Based on the evidence here reported and previously reports demonstrating that ZnMP is required for B. abortus virulence, we can conclude that this postulated TA system is implicated in the regulation of the stress response and virulence of this bacterium"
We hope to make progress towards additional experiments that will allow us to understand the function of this TA system; however, currently we know how a part of this putative TA system participates in the stress response and/or virulence”
Finally, with these changes, we hope to address your concerns regarding our work.
Best regards
Round 2
Reviewer 3 Report
I have no new comments for this revised manuscript, but I really hope you can promote this project with consideration of my previous comments.